

# Distinct metabolic profiling is correlated with bisexual flowers formation resulting from exogenous ethephon induction in melon (*Cucumis melo* L.)

Siyu Fang[1,*], Yaqian Duan[1,*], Lanchun Nie[1,2,3], Wensheng Zhao[1,2,3], Jiahao Wang[1], Jiateng Zhao[1], Liping Zhao[4] and Lei Wang[4]

[1] College of Horticulture, Hebei Agricultural University, Baoding, China
[2] Hebei Key Laboratory of Vegetable Germplasm Innovation and Utilization, Baoding, China
[3] Collaborative Innovation Center of Vegetative Industry of Hebei Province, Baoding, China
[4] Bureau of Agriculture and Rural of Dingzhou, Dingzhou, China
[*] These authors contributed equally to this work.

Corresponding authors
Lanchun Nie, 13784960296@139.com, yynlc@hebau.edu.cn
Wensheng Zhao, zhaowensheng@hebau.edu.cn

## ABSTRACT

Melon (*Cucumis melo* L.) is an agronomically important vegetable. Most cultivars of melon are andromonoecious and bisexual flowers only emerged from the leaf axil of lateral branches. However, the regulatory mechanism contributing to the occurrence of bisexual flowers were still obscure. In this study, ethephon was applied in two common cultivars of melon. In control without ethephon treatment, no bisexual flower was made in the main stem. However, $6.56 \pm 1.42$ and $6.63 \pm 0.55$ bisexual flowers were respectively induced in main stem of 'Yangjiaocui-QX' and 'Lvbao' after ethephon treatment, and induced bisexual flowers distributed in 12–20 nodes of main stem. During the formation of bisexual flowers, 41 metabolites were significantly up-regulated and 98 metabolites were significantly down-regulated. According to the KEGG enrichment analysis of 139 different metabolites, a total of 30 pathways were mapped and KEGG terms of "Phenylalanine, tyrosine and tryptophan biosynthesis", "Phenylalanine metabolism" and "Flavone and flavonol biosynthesis" were significantly enriched. In three significantly enriched KEGG terms, shikimic acid, L-tryptophan, L-phenylalanine, and kaempferol were significantly up-regulated while L-tyrosine, 4-hydroxycinnami acid and luteolin were significantly down-regulated in ET compared to CK. Different metabolites were also classified depend on major class features and 14 classes were acquired. The results of metabonomics and endogenous hormone identification indicated that ethylene could enhance the concentration of salicylic acid, methyl jasmonate, ABA and IAA. This study provided an important theoretical foundation for inducing bisexual flowers in main stem and breeding new varieties of melon in future.

## INTRODUCTION

Melon (*Cucumis melo* L.) is an important horticultural crop and belongs to annual herbs of Cucurbits (*Pitrat, 2016*). Cucurbits in particular cucumber have become good model plants

to study sex differentiation over the past few decades for their abundant kinds of flowers which include male flowers, female flowers and bisexual flowers (*Galun, 1961*; *Switzenberg et al., 2014*; *Boualem et al., 2015*; *Pannell, 2017*; *Pawełkowicz et al., 2019*). Depending on different distributions or combinations of three types of flowers, various sex types appeared including monoecy, andromonoecy, gynomonoecy, gynoecy, hermaphrodite, androecy and trimonoecy in Curcurbitaceae (*Martin et al., 2009*; *Li et al., 2019*). However, the regulatory mechanism of sex differentiation was relatively less in melon than cucumber (*Malepszy & Niemirowicz-Szczytt, 1991*). In melon production, most cultivars are andromonoecious but bisexual flowers only emerged from the leaf axil of lateral branches which is different from cucumber and watermelon because most of them are monoecy and female flowers can arise from the leaf axil either in lateral branches or main stem (*Tanaka et al., 2007*). Therefore, the fruits of melon will only appear in lateral branches and some disadvantages come up with this phenomenon such as increased labors of plant training, undesirable ventilation and light condition, and extended growth and development cycle. Aiming at the above-mentioned issues, it is worthwhile to explore the mystery of sex differentiation and investigate the mechanism for promoting the production of bisexual flowers in main stems of melon.

Sex differentiation of melon can be regulated by multiple environmental factors such as temperature, photoperiod, mineral nutrition, phytohormones or exogenous growth regulators (*Whitaker, 1931*; *Brantley & Warren, 1960*; *Papadopoulou et al., 2005*). Additional male flowers and perfect flowers were produced with the increased concentration of nitrogen under long days but the increase of perfect flowers was proportionately greater, and male flowers and perfect flowers were produced earlier under short days (*Brantley & Warren, 1960*). In spring with low temperatures and short-day photoperiod, female or bisexual flowers are suitable to grow and develop but the femaleness was inhibited with the arrival of high temperatures and long-day photoperiod of summer (*Penaranda et al., 2007*; *Ji et al., 2015*; *Lai, Shen & Zhang, 2018*). After the application of exogenous ethephon, an ethylene releasing agent, female or bisexual flowers in monoecious or andromonoecious genotypes were obviously induced whereas the number of male flowers were significantly decreased in melon (*Zdenka et al., 2013*; *Pitrat, 2016*; *Ye et al., 2020*). However, gibberellic acid reduced the number of female flowers or bisexual flowers and the effect of gibberellin was more visible in andromonoecious plants of melon (*Zdenka et al., 2013*; *Pitrat, 2016*). In addition, both the occurrence of male and bisexual flowers was depressed by 1-Naphthylacetic acid (NAA) under long days, and the reduction of male flowers was proportionately greater. However, NAA treatment promoted flowering and increased the proportion of perfect flowers under short days and there was a decrease in both kinds of flowers with the increased concentrations of NAA in melon (*Brantley & Warren, 1960*). Therefore, the sex differentiation of melon was a combined effect of many external factors.

At present, sex determination of melon was mainly controlled by the *andromonoecious* gene *CmACS-7* (*1-aminocyclopropane-1- carboxylic acid synthase*), the *gynoecious* gene *CmWIP1* (*Wound Inducible Protein1*) and the *androecious* gene *CmACS11*. *CmACS-7* was mainly expressed in carpel primordia of female and bisexual flowers but there

was no accumulation in male flowers, and played an important role in suppressing the development of the stamens in female flowers. In monoecious lines, *CmACS-7* was required for the development of female flowers, whereas bisexual flowers were formed in andromonoecious plants because of the reduced expression of *CmACS-7* (*Boualem et al., 2008*). *CmWIP1* was able to represses the activity of *CmACS-7* to permit the stamen development and the expression of *CmWIP1* resulted in the abortion of carpel which led to the formation of unisexual male flowers (*Martin et al., 2009*). *CmACS11* was strongly expressed in the phloem of female buds and bisexual buds but not in male buds and was needed for the carpel development. *CmACS11* can repress the expression of *CmWIP1* and the plants with loss-of-function of *CmACS11* were transformed into androecy (*Boualem et al., 2015*). In addition, plentiful regulators related to "plant hormone signal transduction", "MAPK signaling pathway" and "carbon metabolism" were found to different expressed during the process of the occurrence of bisexual flowers (*Ge et al., 2020*).

Metabolites play an important role in various stages of plant growth and development, and metabolites are the products of gene regulation (*Fu et al., 2021*). But the different metabolites during the formation of bisexual flowers in melon have never been reported. The plant kingdom has made 100,000–200,000 metabolites which provide a great challenge for scientific research (*Oksman-Caldentey & Inzé, 2004*). Fortunately, metabolomics has been proved to be a meritorious tool for the study of plant development (*Allwood & Goodacre, 2010*). In this study, metabonomic analysis was performed in shoot apex of main stem from ethephon treatment (ET) and control (CK) to investigate the different metabolites contributing the occurrence of bisexual flowers in melon. At the same time, phenotypes after ethephon treatment and levels of several endogenous hormones were surveyed.

## MATERIALS AND METHODS

### Plant materials and growth conditions

Inbred lines 'Yangjiaocui-QX' and 'Lvbao' are two thin skin melon variety and mainly cultivated in Hebei province of China. The seeds of 'Yangjiaocui-QX' and 'Lvbao' were originally obtained from the Hebei Lvlu Agricultural Science and Technology Limited Liability Company (Raoyang, Hengshui, Hebei Province, China) and purified by selfing of six generations by which stably inherited traits were acquired. Both 'Yangjiaocui-QX' and 'Lvbao' are andromonoecy in which bisexual flowers only emerged in lateral branches and used for phenotype investigation after ethephon treatment. 'Yangjiaocui-QX' was also used for metabonomic analysis and endogenous hormones detection. The melon plants were grown in the greenhouse of Hebei Agricultural University in Baoding with the planting space of $70 \times 35$ cm. The temperature was controlled at 25–30 °C/12–15 °C of day/night and normal water management and pest control were implemented.

### Exogenous ethephon treatment

Exogenous ethephon treatment of melon plants were performed as previously described (*Ge et al., 2020*). During the development period of 4–6 internodes, ethephon treatment was performed and 5 mL of 200 ppm ethephon dissolving in water was uniformly sprayed

on the shoot apex of main stem by the atomizer. The plants treated with equivalent water were used for control. Total of fifty plants were treated by ethephon in which thirty and twenty plants were used for phenotypic statistics and metabolome analysis, respectively. The plant height, diameter of stem and length of internode were measured at 15 days after ethephon treatment. The diameter of stem and length of internode were measured at the third internode below the shoot apex of main stem.

## UHPLC- MS analysis of shoot apex

The shoot apex of main stem after removing visible leaves was used for metabolite extraction. Five shoot apexes were mixed as one biological replicate and three biological replicates were performed for ET and CK. The metabolites were extracted as previously described (De Vos et al., 2007; Chen et al., 2013) and stored at −80 °C until the ultrahigh performance liquid chromatography-mass spectrometry (UHPLC-MS) analysis.

The UHPLC separation was implemented by ACQUITY UPLC HSS T3 column (1.8 μm, 2.1 × 100 mm, Waters) with the injection volume of 2 μL. The mobile phase A and B were 0.1% formic acid in water and acetonitrile. The temperature of column and auto-sampler were 40 °C and 4 °C, respectively. The AB Sciex QTOF mass spectrometer was used to acquire high resolution mass spectra data through the information-dependent acquisition (IDA) model (Luo et al., 2016). In this mode, Analyst, a data acquisition software, automatically selects ions and collects its secondary mass spectra data based on the primary mass spectra data and preset standards. The conditions of ESI source were set as following: IonSpray Voltage: +5,500/−4,500 V, Curtain Gas: 35 psi, Temperature: 400 °C, Ion Source Gas 1:60 psi, Ion Source Gas 2:60 psi, DP: ±100 V. Mass spectrometry was performed on multiple-reaction monitoring (MRM) model during the data acquisition (Zha et al., 2018).

## Data analysis

The collected data were analyzed by the Biotree DB database (Biotree, China) and MAPS software (version 1.0). In the first place, the high-resolution MS data were converted to the mzXML format by Proteo Wizard software. Subsequently, a data matrix consisted of the retention time (RT), mass-to-charge ratio (m/z) values, and peak intensity was generated by MAPS software (Smith et al. 2006). Metabolite identification was carried out by the in-house MS2 database and MRM data were processed by the Skyline software (Kuhl et al., 2012). The metabolites with the value of variable importance in the projection (VIP) >1 and $P < 0.05$ (student $t$ test) were assigned to be differentially changed (Saccenti et al., 2014). Kyoto Encyclopedia of Genes and Genomes (KEGG, http://www.genome.jp/kegg/) and MetaboAnalyst (http://www.metaboanalyst.ca/) were used for pathway enrichment analysis (Kanehisa & Goto, 2000; Xia et al., 2015).

## Endogenous hormone identification

To examine the contents of abscisic acid (ABA), indole-3-acetic acid (IAA), $GA_3$ and zeatin riboside (ZR), approximately 0.1 g samples were harvested from 0, 1, 3, 5 days shoot apex of main stem after ethephon treatment and control. Hormone analysis was performed using an

enzyme-linked immunosorbent assay after sample extraction as described (*Konstantinou, 2017*). Three biological replicates were performed for each sample.

## RESULTS

### Ethylene facilitates the production of bisexual flowers in main stem

The phytohormone ethylene was proved to play an important role in sex differentiation of Cucurbitaceae (*Bai & Xu, 2013*; *Manzano et al., 2014*). Ethylene or its releasing agent and inhibitor of ethylene biosynthesis or perception have been widely applied to regulate sex expression in melon, cucumber and watermelon (*Yin & Quinn, 1995*; *Manzano et al., 2014*; *Zhang et al., 2017a* and *Zhang et al., 2017c*). Previous study suggested that ethylene can induce the occurrence of bisexual flowers in melon (*Ge et al., 2020*). In this study, ethephon, an ethylene releasing agent, was also applied to verify the effect of ethylene on the induction of bisexual flowers in two common cultivars of melon.

As expected, evident bisexual flowers were induced in main stem after ethephon treatment in 'Yangjiaocui-QX' and 'Lvbao' (Figs. 1A and 1B, Table 1). Compared to control with no bisexual flowers produced, the numbers of bisexual flowers occurred in main stem of 'Yangjiaocui-QX' and 'Lvbao' with ethephon treatment were $6.56 \pm 1.42$ and $6.63 \pm 0.55$, respectively (Table 1), and induced bisexual flowers distributed in 12–20 nodes of main stem (Figs. 1C and 1D). These observations showed that the effect of ethephon has time-dependent feature and bisexual flowers were visible after 6–8 nodes from ethephon treatment. In addition, effects of ethephon on plant development were investigated. The height of plant and the length of internode were significantly decreased after 15d of ethephon treatment compared to control (Table 1). The diameter of stem was significantly increased upon ethephon treatment (Table 1).

### Identification and analysis of different metabolites in shoot apex of main stem between ethephon treatment and control

To investigate the mechanism for the production of bisexual flowers, the UHPLC-MS analysis was performed in shoot apex of main stem between ethephon treatment and control. Total ion chromatography (TIC) of positive ion mode and negative ion mode of two quality control (QC) samples were shown in Fig. 2. The retention time and peak area of QC samples ($R^2 = 0.991$) were well overlapped which indicated reliable data quality and low instrument error (Fig. 2).

Principle component analysis (PCA) showed that all sample points were located in the Hotelling's T-squared ellipse and sample points of ET can be obviously distinguished from sample points of CK (Fig. 3A). The model parameter of PCA was $R^2X = 0.763$. The score of PC1 and PC2 were 30.9% and 27.4%, respectively (Fig. 3A). These results suggested that distinct metabolic profiling existed between ET and CK and the model can be used for the screening and analysis of different metabolites.

In order to get better visualization and subsequent data analysis, orthogonal projections to latent structures-discriminant analysis (OPLS-DA) were performed to maximize the difference between ET and CK. The model parameters of OPLS-DA were $R^2X = 0.581$, $R^2Y = 1$, $Q^2 = 0.919$ (Fig. 3B). The OPLS-DA results indicated that all the samples were

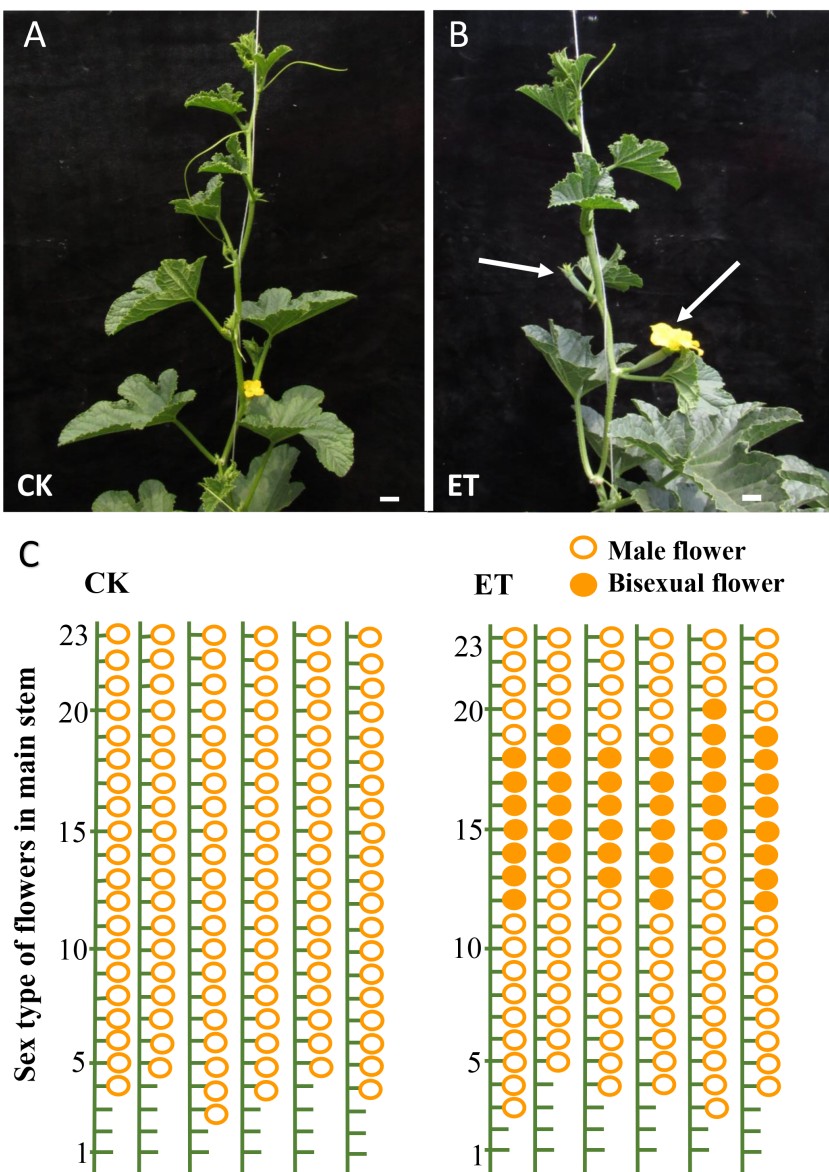

**Figure 1 Phenotypes of 'Yangjiaocui-QX' plants before and after ethephon treatment.** (A) The plants without ethephon treatment. (B) The plants after ethephon treatment. Bisexual flowers (white arrows) were induced in leaf axils of main stem. (C) Diagram of flowers of different sex types on main stem. Induced bisexual flowers (6.56 ± 1.42) distributed in 12–20 nodes of main stem after ethephon treatment. The hollow circles and solid circles represent male flowers and bisexual flowers, respectively. Bars represent two cm in Figs. 1A and 1B.

at a 95% confidence interval and clear separation between the ET and CK were observed (Fig. 3B). Therefore, these models were reliable and could be further applied to screen and analyze the different metabolites.

**Table 1  Effects of ethephon in production of bisexual flowers and growth of plants.**

| Species | The concentration of ethephon treatment | Numbers of plants for phenotypic statistics | Numbers of bisexual flowers in main stem | The height of plant | The diameter of stem | The length of internode |
|---|---|---|---|---|---|---|
| 'Yangjiaocui-QX' | 0 | 30 | 0 | 108.5 ± 5.2a | 5.01 ± 0.34b | 6.88 ± 0.45a |
| | 200 ppm | 30 | 6.56 ± 1.42a | 81.32 ± 10.2b | 5.89 ± 0.55a | 5.46 ± 0.81b |
| 'Lvbao' | 0 | 30 | 0 | 158.42 ± 7.85a | 7.65 ± 0.6a | 9.01 ± 1a |
| | 200 ppm | 30 | 6.63 ± 0.5a | 127.34 ± 8.84b | 7.93 ± 0.53a | 7.65 ± 0.85b |

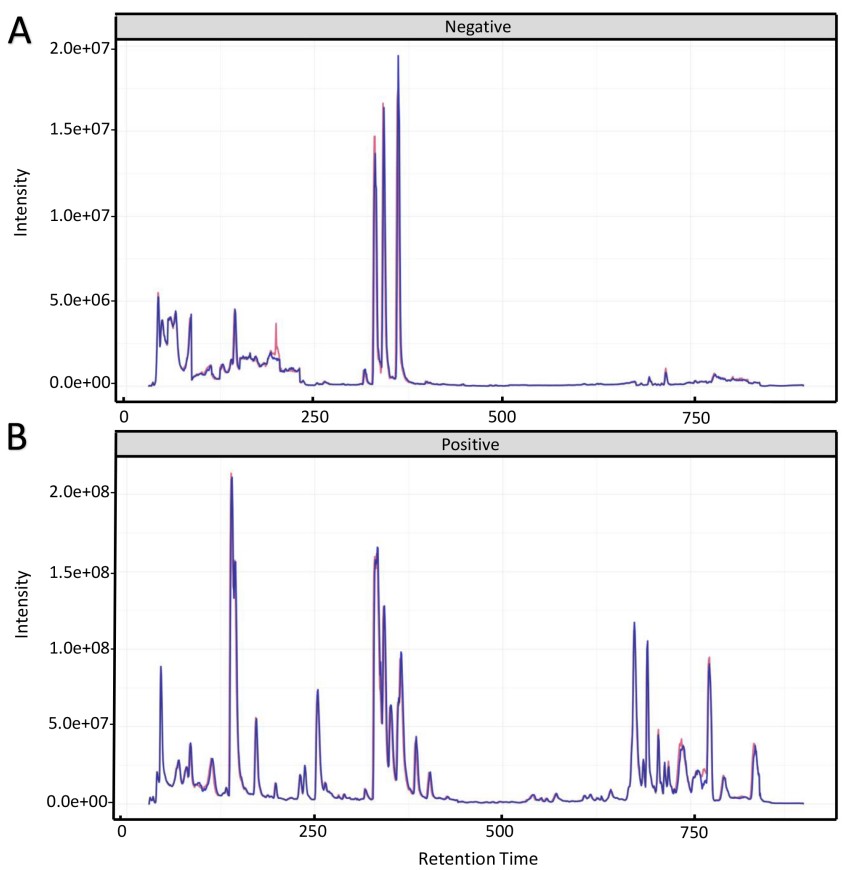

**Figure 2  Total ion chromatography of quality control samples in positive ion mode (A) and negative ion mode (B).** The retention time and peak area of QC samples were well overlapped.

## Different metabolites contribute to the occurrence of bisexual flowers in main stem

The variable importance in the projection (VIP) and $P$-value of Student's $t$-test have been widely used for the identification criterion to screen different metabolites among groups (*Saccenti et al., 2014*). According to the OPLS-DA results, 139 different metabolites were detected between ET and CK using the screening condition of VIP >1 and $P < 0.05$ (Fig. 4). Among these different metabolites, 41 metabolites were significantly up-regulated

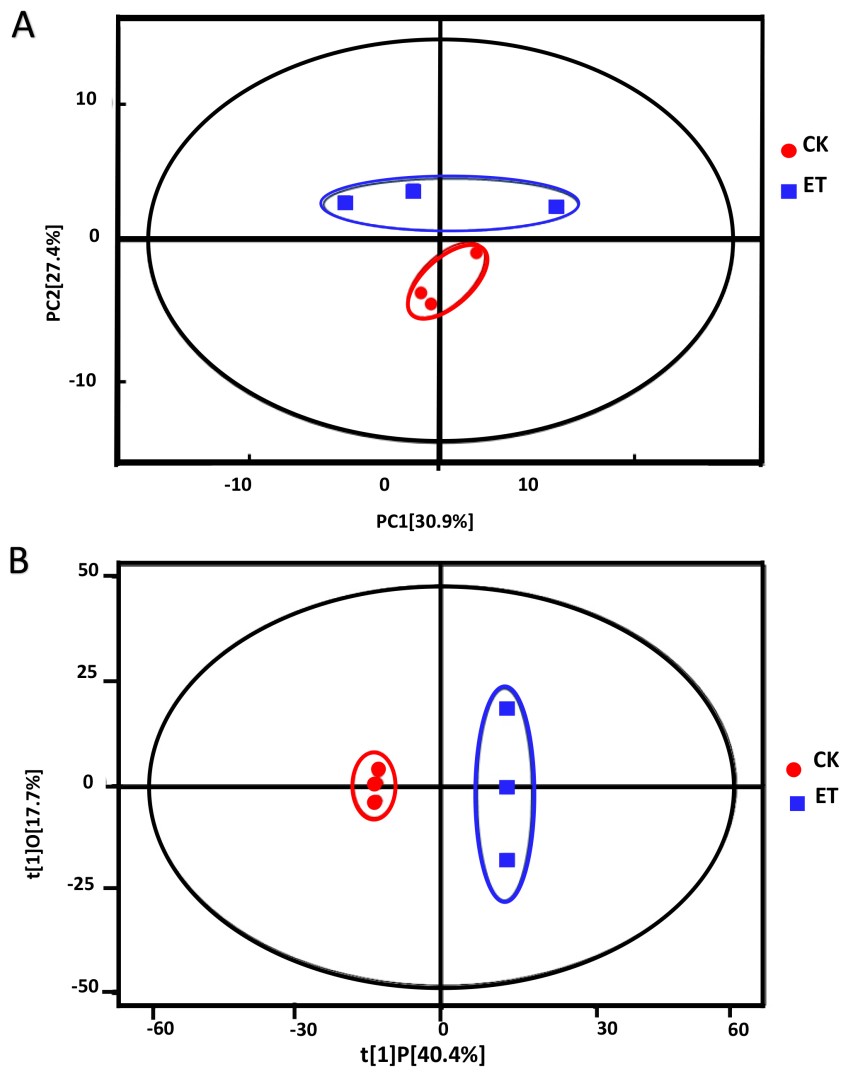

**Figure 3** **Score scatter plots of PCA and OPLS-DA model for group CK *vs* ET.** (A) PCA score plot of CK and ET. $R^2X = 0.763$. (B) OPLS-DA score plot of CK and ET. $R^2X = 0.581$, $R^2Y = 1$, $Q^2 = 0.919$.

and 98 metabolites were significantly down-regulated (Fig. 4A). In addition, hierarchical clustering analysis was performed based on the quantitation of different metabolites. Heat maps showed that up-regulated and down-regulated metabolites were distinctly divided into five and six categories, respectively (Figs. 4B and 4C).

To inquiry the metabolic pathway related to the occurrence of bisexual flowers, Kyoto Encyclopedia of Genes and Genomes (KEGG) term enrichment analysis of different metabolites was performed. The results showed that a total of 30 pathways were mapped and KEGG terms of "Phenylalanine, tyrosine and tryptophan biosynthesis", "Phenylalanine metabolism" and "Flavone and flavonol biosynthesis" were significantly enriched depending on 139 different metabolites (Fig. 5A; Table S1). Furthermore, relative quantification of different metabolites involved in three significantly enriched KEGG terms

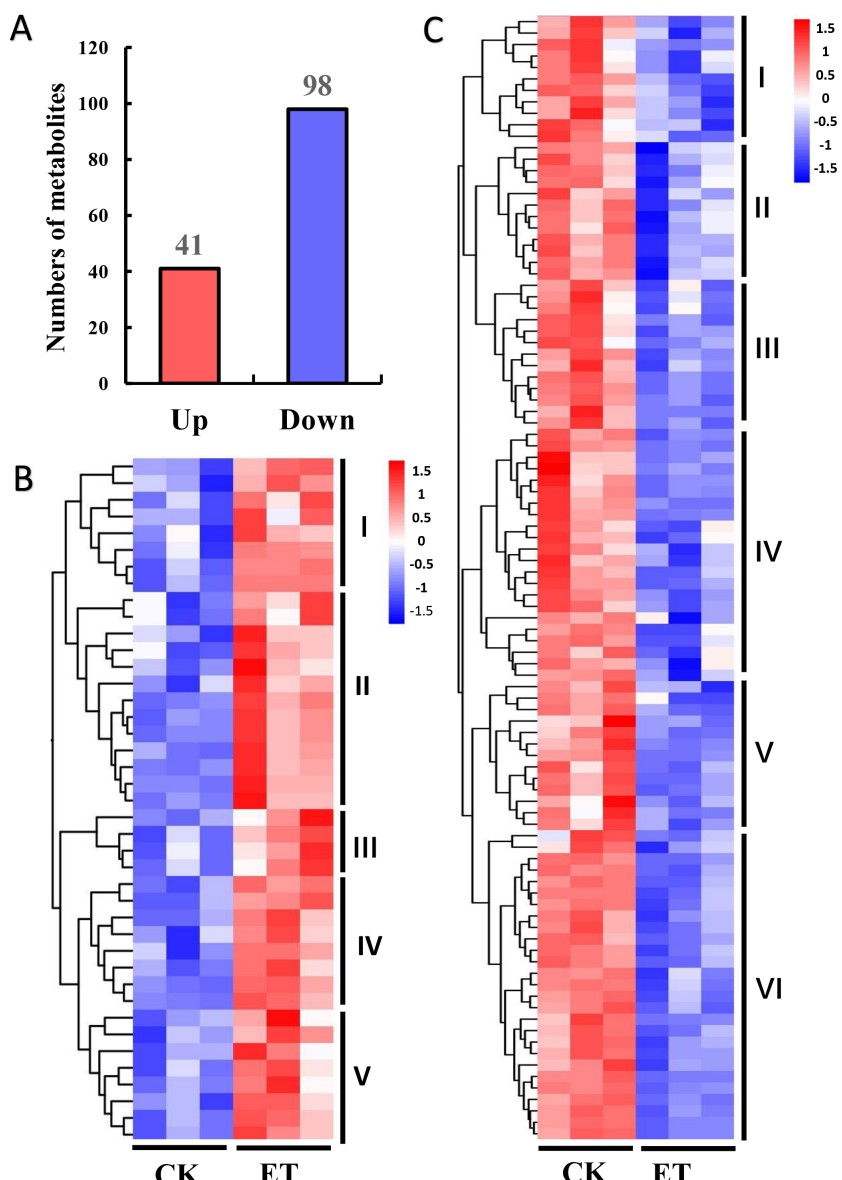

**Figure 4 Hierarchical clustering analysis of differentially expressed metabolites between CK and ET.** (A) Numbers of up-regulated and down-regulated metabolites in ET compared to CK. (B and C) Heatmap of hierarchical clustering analysis for group CK *vs* ET. Up-regulated (B) and down-regulated (C) metabolites were distinctly divided into five and six categories, respectively. The color blocks represent the relative expression of metabolites by which high and low expression were respectively shown in red and blue color.

were investigated. In "Phenylalanine, tyrosine and tryptophan biosynthesis" pathway, four differential metabolites (L-tyrosine, shikimic acid, L-tryptophan and L-phenylalanine) were mapped (Table S1). Shikimic acid, L-tryptophan and L-phenylalanine were significantly up-regulated but L-tyrosine was significantly down-regulated in ET compared to CK (Fig. 5B). The pathway of "Phenylalanine metabolism" also includes 4-hydroxycinnami

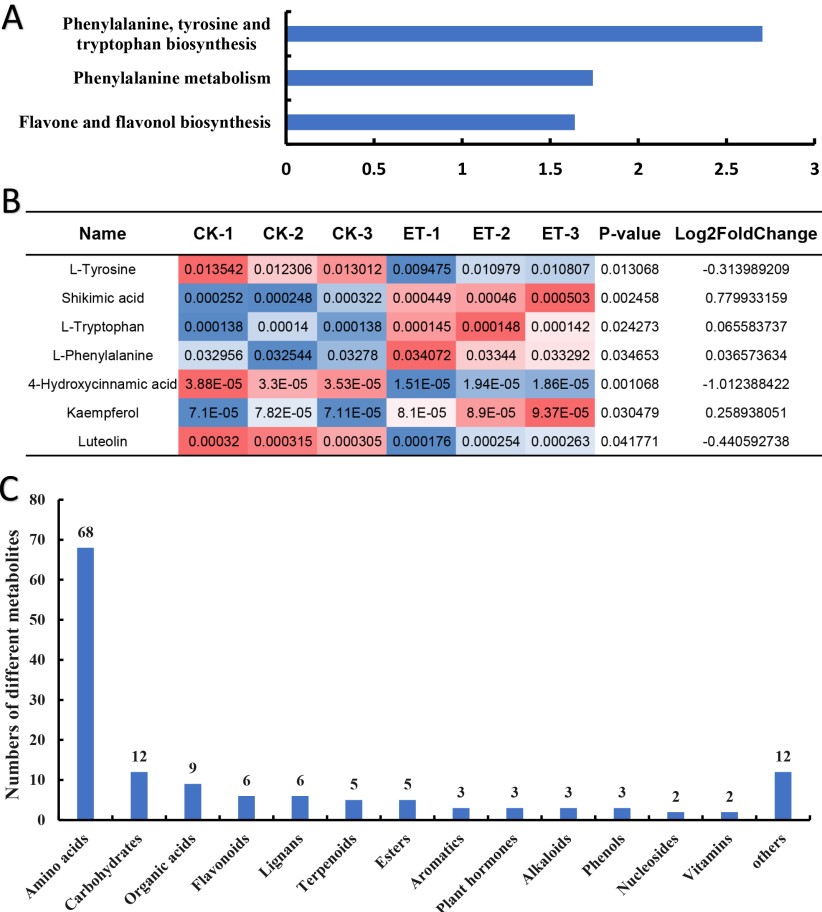

**Figure 5  Classification of different metabolites contributing to the occurrence of bisexual flowers.** (A) KEGG term enrichment analysis of different metabolites. "Phenylalanine, tyrosine and tryptophan biosynthesis", "Phenylalanine metabolism" and "Flavone and flavonol biosynthesis" pathways were significantly enriched. (B) The quantification of different metabolites involved in three significantly enriched KEGG terms. Significantly up-regulated metabolites included shikimic acid, L-tryptophan, L-phenylalanine, and kaempferol. Significantly down-regulated metabolites included L-tyrosine, 4-hydroxycinnami acid and luteolin. The red color indicated higher content while blue indicated lower content of different metabolites. (C) Classification of 139 different metabolites. A total of 14 classes were acquired including 68 amino acids or mini peptide, 12 carbohydrates, nine organic acids, six flavonoids, six lignans, five terpenoids, five ester, three aromatics, three plant hormones, three alkaloids, three phenols, two nucleosides, two vitamins and 12 other compounds.

acid besides above-mentioned L-phenylalanine but the content of 4-hydroxycinnami acid was significantly decreased in ET (Fig. 5B; Table S1). In pathway of "Flavone and flavonol biosynthesis", kaempferol was significantly up-regulated while luteolin was significantly down-regulated in ET compared to CK (Fig. 5B; Table S1). These results suggested that "Phenylalanine, tyrosine and tryptophan biosynthesis", "Phenylalanine metabolism" and "Flavone and flavonol biosynthesis" were related to the occurrence of bisexual flowers in melon.

In addition, different metabolites were also classified depend on major class features. According to the variety of 139 different metabolites, 14 classes were acquired which include 68 amino acids or mini peptide, 12 carbohydrates, nine organic acids, six flavonoids, six lignans, five terpenoids, five ester, three aromatics, three plant hormones, three alkaloids, three phenols, two nucleosides, two vitamins and 12 other compounds (Fig. 5C; Table S2). As shown in Table S2, most amino acids or mini peptide (62) were significantly down-regulated while only six amino acids or mini peptide were significantly up-regulated. Consistent with involved metabolic pathway of "Flavone and flavonol biosynthesis", four flavonoids were significantly increased and two flavonoids were significantly decreased in shoot apex of ET compared to CK (Table S2). However, the largest number of different metabolites were carbohydrates except for amino acids or mini peptide. Seven carbohydrates were significantly up-regulated and five carbohydrates were significantly down-regulated in comparison of ET to CK (Table S2). Among the up-regulated carbohydrates, the fold change of "Maltotriose" reached 3.38 with the $p$-value of 0.006 (Table S2). In addition, multiple organic acids were significantly changed with five up-regulation and four down-regulation (Table S2).

### Ethylene interacts with other endogenous hormones to regulated the formation of bisexual flowers

In shoot apex of main stem after ethephon treatment, the content of salicylic acid (SA) and methyl jasmonate (MeJA) were significantly increased but ($\pm$)7-epi jasmonic acid was significantly down-regulated (Table S2). We also detect the content variation of ABA, IAA, GA$_3$ and ZR in shoot apex of main stem at 0,1,3,5 days after ethephon treatment to investigate the effect of ethephon on other endogenous hormones. In shoot apex of CK, the content of all detected hormones showed no appreciable difference (Fig. 6). In shoot apex of ET, the ABA concentration exhibited a significant peak at 1 day after treatment (DAT) and gradually decreased with the increase of treatment days (Fig. 6A). Similar to ABA, the content of IAA was also significantly increased in ET but the peak of IAA was at 3 DAT (Fig. 6B). The content of ABA and IAA in ET approach to CK at 5 DAT (Figs. 6A and 6B). However, the concentration of GA$_3$ and ZR displayed no significant difference after ethephon treatment (Figs. 6C and 6D). These results indicated that ethylene could enhance the concentration of SA, MeJA, ABA, IAA during the formation of bisexual flowers.

## DISCUSSION

Most cultivars of melon are andromonoecious but bisexual flowers only emerged from the leaf axils of lateral branches (*Tanaka et al., 2007*). Ethylene plays an important role in sex differentiation and can effectively enhance the proportion of female or bisexual flowers in cucurbits (*Iwahori, Lyons & Sims, 1969*; *Müller & Munné-Bosch, 2015*; *Van de Poel, Smet & Van Der Straeten, 2015*; *Zhang et al., 2017a*; *Zhang et al., 2017c*; *Wang et al., 2019*). In this study, bisexual flowers emerged from the leaf axils of main stem after ethephon treatment in two melon cultivars ('Yangjiaocui-QX' and 'Lvbao', Fig. 1B, Table 1) and induced bisexual flowers mainly distributed in 12–20 nodes of main stem (Fig. 1D). Previous studies showed that nine transcription factors including one ETHYLENE RESPONSE1

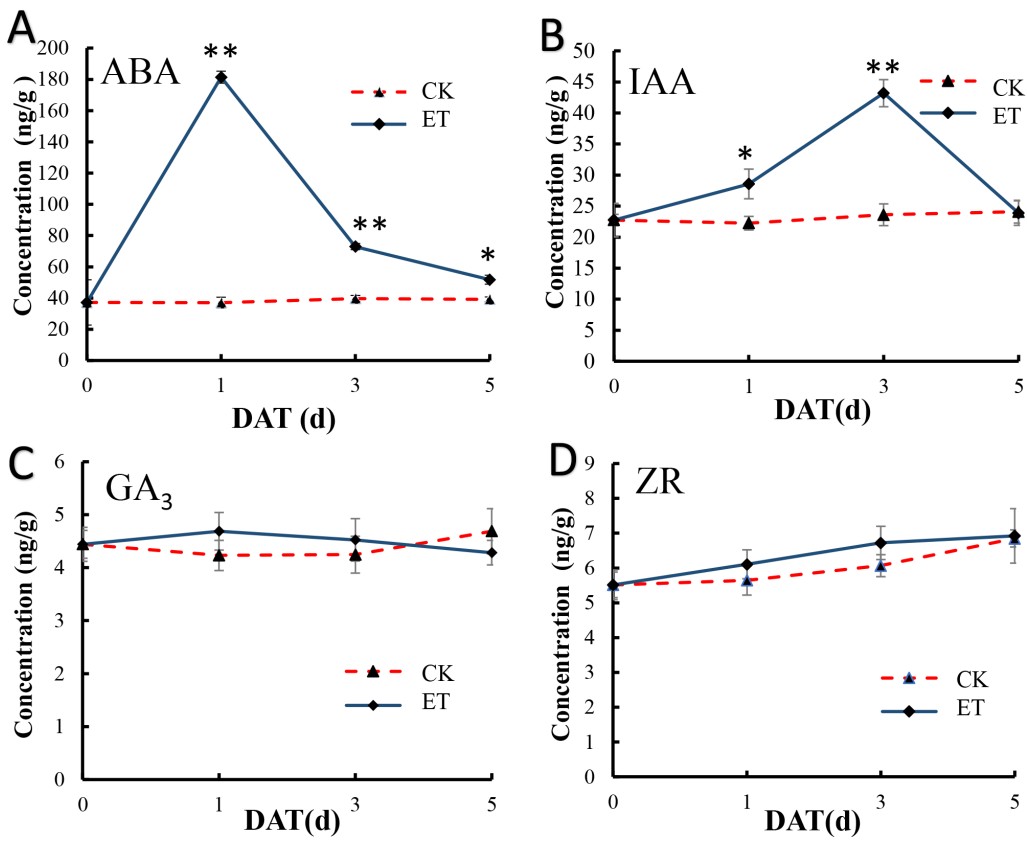

**Figure 6 Hormone measurements in ET and CK.** The ABA (A) and IAA (B) concentration exhibited a significant peak at 1 DAT and 3 DAT in ET compared to CK. The concentration of GA$_3$ (C) and ZR (D) displayed no significant difference between ET and CK.

(ETR1), two ETHYLENE INSENSITIVE 3 (EIN3), three EIN3-binding F-box proteins (EBFs) and three ETHYLENE RESPONSE FACTORs (ERFs) involved in ethylene signaling pathway were found to significantly up-regulated upon ethephon induction in shoot apex of main stem (Fig. 7B, Table S3) (*Ge et al., 2020*). Above-mentioned facts suggested that bisexual flowers were induced with the increased content of ethylene and the effect of ethephon was not affected by the applied location.

In addition, the concentration of SA, MeJA, ABA and IAA were significantly up-regulated after ethephon treatment (Figs. 6A and 6B), and either ABA or IAA had been proved to be the key players in inducing femaleness (*McAdam et al., 2016*; *Li et al., 2019*; *Wang et al., 2019*). While the application of GA$_3$ can markedly increase the number of male flowers (*Zhang et al., 2017c*) and there was no significant difference acquired in GA$_3$ content after ethephon treatment in this study (Fig. 6C). These results indicated that ethylene may promote the occurrence of bisexual flowers by interacting with these elevated hormones.

Metabolites act as the important regulator for plant phenotype (*Fu et al., 2021*). In this study, 139 different metabolites were obtained between ET and CK by UHPLC-MS and three significantly enriched pathways were mapped by KEGG analysis

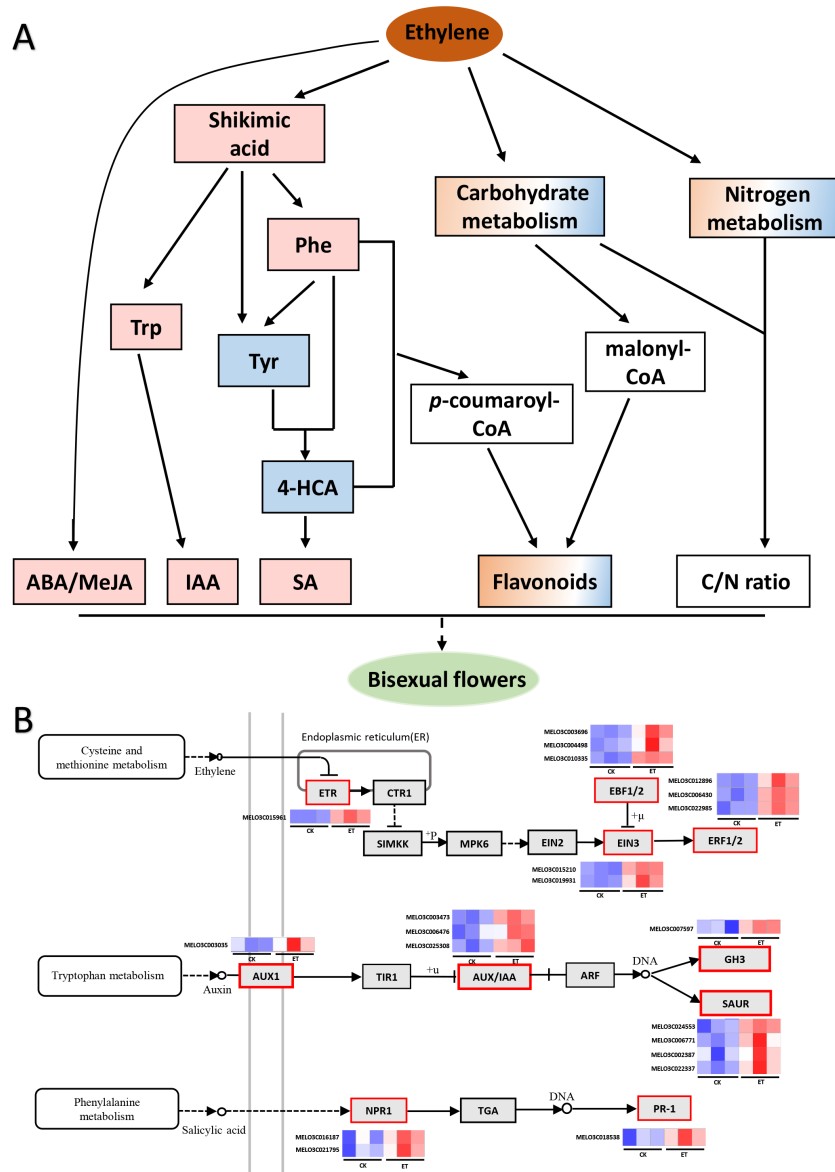

**Figure 7  Working model for the occurrence of bisexual flowers and heat maps of DEGs involved in ethylene, auxin and salicylic acid signaling pathways.** (A) Working model for the occurrence of bisexual flowers. After the application of ethephon, ABA and MeJA were significantly induced by an unknown pathway. Shikimic acid was significantly up-regulated and then form three essential aromatic amino acids including tryptophan, phenylalanine and tyrosine. Subsequently, IAA was biosynthesized depend on tryptophan metabolism. L-phenylalanine and tyrosine were able to form 4-hydroxycinnamic acid, and then to synthesize SA. L-phenylalanine and 4-hydroxycinnami acid also participate in phenylpropanoid biosynthesis pathway and generated *p*-coumaroyl-CoA can act as the precursors to synthesize many flavonoids. Besides *p*-coumaroyl-CoA, flavonoids can also be synthesized by the precursor malonyl-CoA which derived from carbohydrate metabolism. In addition, metabolites involved in carbon metabolism and nitrogen metabolism were differently expressed which led to the change of C/N ratio. The occurrence of bisexual flowers resulted from the interaction of multiple factors in melon. 

**Figure 7 (...continued)**
The pink and blue rectangle respectively represent up-regulated and down-regulated metabolites. The gradient rectangle represents multi-metabolites including up-regulated and down-regulated metabolites. The hollow rectangle represents the unknown variation. (B) Heat maps of differentially expressed genes involved in ethylene, auxin and salicylic acid signaling pathways. The blue color indicated lower expression while red indicated higher expression. The transcriptome data were acquired in Sequence Read Archive (SRA) database of the National Center for Biotechnology Information (https://www.ncbi.nlm.nih.gov/) with accession number PRJNA657708.

(Figs. 4 and 5). Among three significantly enriched pathways, shikimic acid was the major reason that "Phenylalanine, tyrosine and tryptophan biosynthesis" and "Phenylalanine metabolism" pathways were mapped. In higher plants, shikimic acid can be metabolized into chorismic acid, and then form three essential aromatic amino acids including tryptophan, phenylalanine and tyrosine (Fig. 7A) (*Weaver & Herrmann, 1997*; *Wilson et al., 1998*; *Mir, Jallu & Singh, 2015*). On the one hand, indoleacetic acid was biosynthesized depend on tryptophan metabolism (*Nigović et al., 2000*). On the other hand, phenylalanine can be further hydroxylated to tyrosine, and these two amino acids are catalyzed by phenylalanine ammonia-lyase (PAL) to form cinnamic acid and hydroxycinnamic acid, and then to synthesize the salicylic acid (Fig. 7A) (*Wang et al., 2005*; *Ding & Ding, 2020*; *Lynch et al., 2020*).

Compared to CK, nine differentially expressed genes (DEGs) in "Tryptophan metabolism", nineteen DEGs in "Phenylalanine metabolism" and eleven DEGs in "Phenylalanine, tyrosine and tryptophan biosynthesis" were found in ET (Table S3) (*Ge et al., 2020*). In this study, the content of shikimic acid, L-tryptophan and L-phenylalanine were significantly up-regulated so that the concentration of IAA and SA were significantly increased after ethephon treatment (Figs. 5B and 6B; Table S2). The reason for decrease of L-tyrosine and 4-hydroxycinnami acid was that these two amino acids may be used to synthesize SA or other unknown metabolites (Fig. 5B). The increased content of IAA led to the significantly up-regulation of one auxin influx carrier AUX1, three Aux/IAA transcriptional repressors, one IAA synthetase GH3 and four SMALL AUXIN UP-REGULATED RNAs (SAURs) in auxin signaling pathway (Fig. 7B, Table S3) (*Ge et al., 2020*). Induced SA resulted in higher transcription accumulation of two SA receptors NONEXPRESSER OF PR GENES1 (NPR1) and one pathogenesis-related protein 1 (PR 1) in salicylic acid signaling pathway (Fig. 7B, Table S3) (*Ge et al., 2020*). Future studies using transgenic system such as CRISPR-Cas to obtain transgenic lines of key genes in IAA or SA signaling pathways such as SAURs or NPR1 would be promising to clarify the mechanism in the occurrence of bisexual flowers by IAA and SA in melon. In addition, there were eight and two genes were significantly increased after ethephon treatment in ABA and MeJA signaling pathways, respectively (Table S3) (*Ge et al., 2020*). However, the mechanism that ethylene regulated the synthesis of shikimic acid and promoted the up-regulation of ABA and MeJA need to further studies.

Furthermore, L-phenylalanine and 4-hydroxycinnami acid also participate in phenylpropanoid biosynthesis pathway and generated *p*-coumaroyl-CoA can act as the precursors to synthesize many flavonoids which function as pigments in flowers and

fruits or involved in UV-scavenging, fertility and disease resistance, and can be mainly divided into flavone, flavonol, isoflavone, flavanone, anthocyanidin and so on (Fig. 7A) (*Dooner, Robbins & Jorgensen, 1991*; *Koes, Quattrocchio & Mol, 1994*; *Schijlen et al., 2004*; *Mageney, Neugart & Albach, 2017*). Just as expected, five genes involved in "Flavonoids biosynthesis" were significantly increased (*Ge et al., 2020*) and the "Flavone and flavonol biosynthesis" metabolic pathway was significantly enriched with an up-regulated flavonol (kaempferol) and a down-regulated flavone (luteolin) in ET compared to CK (Figs. 5A and 5B). Moreover, another three flavonoids (orientin 2″-rhamnoside, apigenin 7-cellobioside-4′-glucoside, cyanidin 3-glucoside) were significantly increased and 5,7-dihydroxy-2-(4-hydroxyphenyl)-3,4-dihydro-2H-1-benzopyran-4-one was significantly decreased in shoot apex of ET compared to CK (Table S2). These results indicated that flavonoids were related to the formation of bisexual flowers in melon.

Besides *p*-coumaroyl-CoA, flavonoids can also be synthesized by the precursor malonyl-CoA which derived from carbohydrate metabolism (Fig. 7A) (*Schijlen et al., 2004*). In this study, seven and five carbohydrates were found to significantly increased and decreased in ET compared to CK, respectively (Table S2). Carbohydrates were not only used to synthesize flavonoids but also to provide energy and act as structural material for maintaining normal life activities of plants (*Chiariello, Mooney & Williams, 2000*; *Gibson, Laby & Kim, 2001*). For example, the contents of maltotriose, lactose and stachyose tetrahydrate (fold change = 3.38, 1.82, 1.62, respectively) were significantly increased after ethephon treatment (Table S2). In addition, carbon metabolism-related metabolic pathways such as "Citrate cycle (TCA cycle)", "Glycolysis or Gluconeogenesis" and "Starch and sucrose metabolism" were mapped though there were no significantly difference (Table S1) (*Fernie, Carrari & Sweetlove, 2004*). These results suggested that soluble sugar and energy may be the basis for the formation and development of bisexual flowers.

Additionally, carbon metabolism and nitrogen metabolism are closely interlinked in almost all stages of plant growth and development (*Sakr et al., 2018*). The coordination and interaction of carbon and nitrogen metabolism are essential for floral bud differentiation. Traditional theories considered that high and low C/N ratio will promote and suppress floral bud differentiation, respectively (*Funk, Glenwinkel & Sack, 2013*; *Zhang et al., 2017b*). Compared to CK, fifty-two and nine genes were differently expressed in "Carbon metabolism" and "Nitrogen metabolism" pathways after ethephon treatment, respectively (Table S3) (*Ge et al., 2020*). In this study, the content of L-glutamine was significantly decreased while the content of L-phenylalanine was significantly increased, and these two amino acids were found to involve in nitrogen metabolism (Tables S1 and S2). Furthermore, there still were six up-regulated amino acids though most amino acids or mini peptides (62) were significantly down-regulated (Table S2). Similarly, both up-regulated and down-regulated carbohydrates existed in ET compared to CK (Table S2). These results suggested that carbon metabolites and nitrogen metabolites were also correlated with the occurrence of bisexual flowers in melon (Fig. 7A).

Previous studies indicated that ethylene can increase the number of female flowers or bisexual flowers in melon (*Li et al., 2019*). However, the position and period of ethephon application were obscure. The observations of this study showed that exogenous ethephon

applied in shoot apex can induce the formation of bisexual flowers and the effect of ethephon has time-dependent feature that is bisexual flowers were visible after 6–8 nodes from ethephon treatment and also constantly occurred 6–8 nodes (Fig. 1). Therefore, these results are convenient to melon production for early-maturing cultivation and saving of labor.

## CONCLUSIONS

Exogenous ethephon applied in shoot apex can induce the formation of bisexual flowers and bisexual flowers were visible after 6–8 nodes from ethephon treatment. During the occurrence of bisexual flowers, distinct metabolic profiling existed between ET and CK. A total of 139 metabolites were identified and three KEGG terms ("Phenylalanine, tyrosine and tryptophan biosynthesis", "Phenylalanine metabolism" and "Flavone and flavonol biosynthesis") were significantly enriched during the occurrence of bisexual flowers in melon. Due to the variation of some amino acids and organic acids, the content of endogenous hormones such as SA and IAA were significantly up-regulated. In addition, flavonoids, carbon metabolites and nitrogen metabolites were also related to the occurrence of bisexual flowers in melon.

## ACKNOWLEDGEMENTS

The authors appreciate the members of Nie and Zhao labs for technical assistance and discussions.

### Funding

This study was financially supported by the National Natural Science Foundation of China (32002063), the Natural Science Foundation of Hebei Province (C2021204033), the earmarked fund for Hebei Vegetables Innovation Team of Modern Agro-industry Technology Research System (HBCT2018030210), the Special Research Foundation of Hebei Agricultural University (YJ201837), and the Basic Scientific Research Foundation of Hebei Agricultural University (KY202005). The funders had no role in study design, data collection and analysis, decision to publish, or preparation of the manuscript.

### Grant Disclosures

The following grant information was disclosed by the authors:
National Natural Science Foundation of China: 32002063.
Natural Science Foundation of Hebei Province: C2021204033.
Hebei Vegetables Innovation Team of Modern Agro-industry Technology Research System: HBCT2018030210.
Special Research Foundation of Hebei Agricultural University: YJ201837.
Basic Scientific Research Foundation of Hebei Agricultural University: KY202005.

## Competing Interests

The authors declare there are no competing interests.

## Author Contributions

- Siyu Fang and Yaqian Duan performed the experiments, prepared figures and/or tables, and approved the final draft.
- Lanchun Nie and Wensheng Zhao conceived and designed the experiments, prepared figures and/or tables, and approved the final draft.
- Jiahao Wang, Jiateng Zhao, Liping Zhao and Lei Wang analyzed the data, authored or reviewed drafts of the paper, and approved the final draft.

## Data Availability

The sequencing data is available at MetaboLights (https://www.ebi.ac.uk/metabolights) of the European Molecular Biology Laboratory—European Bioinformatics Institute (EMBL-EBI): MTBLS2993.

## Supplemental Information

Supplemental information for this article can be found online at http://dx.doi.org/10.7717/peerj.13088#supplemental-information.

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
