# Peer review of "Distinct metabolic profiling is correlated with bisexual flowers formation resulting from exogenous ethephon induction in melon (Cucumis melo L.)"

_PeerJ, doi:10.7717/peerj.13088_

## Round 0.1 · original submission · Major Revisions

Please address the comments of the reviewers, in particular:

1) appropriate statistical analysis of differential metabolites.

2) additional support for your derived hypotheses; such as integration of transcriptomic data.

3) publication of raw data, e.g. on Zenodo

Reviewer 1 ·

Basic reporting

General comments:

In the submitted manuscript, the authors described the metabolimc analysis in melon and more particulary in shoot apex of main stem from ethephon treatment and control plants to investigate the different metabolites contributing the occurrence of bisexual flowers.

The authors conducted extensively metabolomic analyses, combined with the experimental validation of phenotypes after etephone treatment. The strong points of the manuscript is a very interesting topic regarding to plant reproduction. The description of pathways are very interesting especially with regard to hormone signaling pathway connections.

Experimental design

The current guidelines for metabolomic analysis are that experiments using metabolome profiling data to describe changes should contain at leat three the biological replicates (preferably is more) and also three technical replicates. Herein authors have only biological replicates, which are bulked. According to the standard protocol, it must contain three biological and three technical replicates. Herein there is lack of this information and thus it seems the analysis are not properly replicated. In materials and methods section – Materials and methods section - p.10 line 139 Five shoot apexes were mixed as one biological replicate and three biological replicates were performed for ET and CK. Please look at the technical quiadance for recommendations for metabolomic profiling data: Fernie AR, Aharoni A, Willmitzer L, et al. Recommendations for reporting metabolite data. Plant Cell. 2011;23(7):2477-2482. doi:10.1105/tpc.111.08627

Validity of the findings

Has statistical reference been made to all identified metabolites? An example is the Bonferoni correction, which adjusts the observed significance level to the fact that many comparisons are made.

The diagram presented by the authors at the end of the article and the conclusions do not have much support in the results described.

·

Basic reporting

This is seems to be good article related to gene expression for flower development in lateral and main axis in melon

Experimental design

Appreciate and carried out properly

Validity of the findings

Findings are seems to new and novel.

Additional comments

Probably it is a new article for sex expression in relation to main axis and lateral branches in melon.

Reviewer 3 ·

Basic reporting

This paper mainly reports metabolic profiling contributes to the formation of bisexual flowers on main stem. By measuring metabolomics and endogenous hormone between main stem with or without ethephon treatment groups, the authors finally identified three significantly enriched KEGG terms participating in the occurrence of bisexual flowers in melon. The endogenous hormones SA and IAA were up regulated by ethephon, might be due to amino acids and organic acids up regulated.
However, there are some major concerns. The most important one is that these are only speculations how substances regulate the formation of hermaphrodite flowers, lacking profound evidence at the molecular level. Ge et al. (2020) have shown the transcriptome data of main stem with and without ethephon treatment. I recommend the authors use the strategy of metabolome and transcriptome joint analysis to construct co-expression network that provides comprehensive information on the dynamics of major metabolites and the transcriptome profiles that underlie them during melon bisexual flower development. By analyzing the co-expression network, the authors could explain metabolites variation from the level of gene expression in bisexual flowers development. It will be more powerful to strengthen the key points that ethylene regulates bisexual flowers development by crosstalking with SA and IAA through regulating some amino acids and organic acids.

Experimental design

This paper provides a potential perspective that metabolites play important role in regulating the bisexual flower development.
However, the title “Distinct metabolic profiling contributes to the formation difference of bisexual flowers between main stem and lateral branches in melon (Cucumis melo L.)” of this article is not appropriate, because in the author's experimental design, only the main stem was treated with ethylene, and the lateral stem group was not set. Both main stem treated with ethylene and lateral stem produce bisexual flowers, and the overlap factors between them should be the causal factors of bisexual flowers formation. Thus, according to this title, lateral stem should be used as another experimental group, except for main stem and main stem treated with ethylene.

Validity of the findings

2. In Figure 5, it seems that the differences of contents of metabolites such as shikimic acid, L-tryptophan, L-phenylalanine was not significant, and the raw data should be provided to confirm the conclusion. Also, the number of biological duplicates should be added.

Additional comments

3. In line 94, there is a spelling mistake that “stamina” should be replaced by “stamens”.

---

## Round 0.2 · Minor Revisions

Dear Authors,

Thank you very much for your detailed response, and for making your data available to the community.

A Section Editor made some additional comments you should address prior to acceptance of your paper:

> "The title is misleading. The authors have identified metabolites that change in the apex in response to ethephon treatment. The resulting changes in metabolites correlate with bisexual flower formation, but they may or may not actually be related to the mechanisms of bisexual flower formation (except for those cases where there is additional experimental evidence, such as IAA). The title, discussion, and conclusion need to be re-worded with this in mind.
>
> There are numerous typos, including "andromonoecies" , "creative flowers", "ether" (instead of "either"), etc.
>
> In the abstract, the fact that no bisexual flower is made in the control should be stated."

Best, Robert

·

Basic reporting

The manuscript is clear and English language seems to be good in thorough out the manuscript. Structure of the article also seems to be as per the standard of this journal.

Experimental design

Sufficient

Validity of the findings

seems to be novel and certainly have academic significant.

Additional comments

This manuscript has been well written and corrections have been incorporated,

---

## Round 0.3 · accepted · Accept

Thank you very much for the additional corrections to your manuscript.